# CC Genotype of *GNAS* c.393C>T (rs7121) Polymorphism Has a Protective Effect against Development of BK Viremia and BKV-Associated Nephropathy after Renal Transplant

**DOI:** 10.3390/pathogens11101138

**Published:** 2022-10-01

**Authors:** Tobias Peitz, Birte Möhlendick, Ute Eisenberger, Winfried Siffert, Falko Markus Heinemann, Andreas Kribben, Justa Friebus-Kardash

**Affiliations:** 1Department of Nephrology, University Hospital Essen, University of Duisburg-Essen, 45147 Essen, Germany; 2Institute of Pharmacogenetics, University Hospital Essen, University of Duisburg-Essen, 45147 Essen, Germany; 3Institute for Transfusion Medicine, Transplantation Diagnostics, University Hospital Essen, University of Duisburg-Essen, 45147 Essen, Germany

**Keywords:** *GNAS* c.393C>T polymorphism, renal transplantation, BK viremia, BKV-associated nephropathy, de novo donor specific antibodies

## Abstract

The *GNAS* gene encodes the alpha-subunit of the stimulatory G-protein (Gαs) in humans and mice. The single-nucleotide polymorphism of *GNAS*, c.393C>T, is associated with an elevated production of Gαs and an increased formation of cyclic adenosine monophosphate (cAMP). In the present study, we analyzed the effect of this *GNAS* polymorphism on a renal allograft outcome. We screened a cohort of 436 renal allograft recipients, who were retrospectively followed up for up to 5 years after transplant. *GNAS* genotypes were determined with polymerase chain reaction restriction fragment length polymorphism (PCR-RFLP) assays. The 393T allele was detected in 319 (73%) recipients (113 recipients with TT and 206 with CT genotype) and the CC genotype in 117 (27%). The CC genotype was associated with a significantly lower frequency of BK viremia (CC, 17 recipients (15%); T 84 (26%)); *p* = 0.01; TT, 27 vs. CC, 17, *p* = 0.07; TT, 27 vs. CT, 57, *p* = 0. 46; CT, 57 vs. CC, 17, *p* = 0.01) and BKV-associated nephropathy (CC, 3 recipients (3%); T, 27 (8%); *p* = 0.03; TT,10 vs. CC, 3, *p* = 0.04; TT, 10 vs. CT,17, *p* = 0.85; CT, 17 vs. CC,3, *p* = 0.04) after transplant. BKV-associated nephropathy-free survival was significantly better among CC genotype carriers than among T allele carriers (*p* = 0.043; TT vs. CC, *p* = 0.03; CT vs. CC, *p* = 0.04; TT vs. CT, *p* = 0.83). Multivariate analysis indicated an independent protective effect of the CC genotype against the development of both BK viremia (relative risk. 0.54; *p* = 0.04) and BKV-associated nephropathy after renal transplant (relative risk. 0.27; *p* = 0.036). The *GNAS* 393 CC genotype seems to protect renal allograft recipients against the development of BK viremia and BKV-associated nephropathy.

## 1. Introduction

The human *GNAS* gene is located on chromosome 20q13.32 and encodes for the G-protein α-subunit (Gαs) [1]. Together with the β- and γ-subunits, the α-subunit forms the heterotrimeric transmembrane G-protein complex. Binding of guanosine diphosphate activates the Gαs and leads to its dissociation from the βγ-complex [2]. Consequently, Gαs stimulates the activity of adenylyl cyclase, resulting in the production of cyclic adenosine monophosphate (cAMP). As a main second messenger, cAMP interacts with cAMP-dependent protein kinases that are involved in diverse intracellular signal cascades, and this interaction causes activation or deactivation of phosphorylated proteins, depending on the cell type [3,4]. G-protein-coupled receptors can be activated by neurotransmitters, chemokines, and hormones. Therefore, G-protein-mediated signaling is involved in numerous signal transduction pathways in all organs and tissues [4].

The silent single-nucleotide polymorphism of the *GNAS* gene, rs7121 (c.393C>T), causes altered mRNA folding and influences mRNA stability, an effect that results in increased expression of mRNA and Gαs protein in persons with the TT/CT genotype [5,6,7]. As a consequence, the expression of activated Gαs and the synthesis of cAMP are lower among carriers of the CC genotype than among carriers of the T allele [6,7]. In the European population, the frequency of the T allele in the *GNAS* c.393C>T polymorphism is approximately 47% [8]. The presence of the T allele has been linked to increased survival rates among patients with various types of solid cancer [8]. In a broad range of cancer entities such as bladder cancer, colorectal cancer, gastric cancer, and non-small lung cancer the appearance of the T allele was related to the decreased rate of cancer progression, reduced occurrence of metastasis, and improved overall survival [6,7,9,10,11]. In contrast, in other selected cancer types such as breast cancer, intrahepatic cholangiocarcinoma, and esophageal cancer TT/CT genotypes exhibit a disadvantageous effect in terms of therapy response. Patients with the CC genotype respond better to chemotherapy or radiation and demonstrate slower cancer progression of the three above-mentioned human cancer entities [8,12,13,14,15]. Moreover, genotypes of the *GNAS* c.393C>T polymorphism are associated with increased risks of essential arterial hypertension, ventricular tachyarrhythmia, sudden cardiac death, and reduced response to beta-blocker therapy [16,17].

Although Gαs-mediated signaling participates in a number of pathophysiological processes by initiating diverse intracellular signaling cascades, until now no studies have elucidated the role of the functional *GNAS* c.393C>T polymorphism in the clinical outcome of renal transplant recipients. However, increasing evidence suggests that inherited factors may affect renal allograft survival [18,19,20,21]. Regarding the relationship with the renal allograft outcome, G-protein-mediated receptor signaling is a promising candidate because it is a part of angiotensin II, chemokine, and adrenergic receptor signaling, which is involved in the immunological processes of the formation of donor-specific antibodies (DSAs) and the development of rejections [4,22,23]. Additionally, the previously described association of *GNAS* genotypes with endothelial dysfunction and vascular hypertrophy may be relevant for chronic allograft dysfunction accelerating allograft loss [4,16]. The Gαs protein encoded by *GNAS* participates in a signaling of numerous receptors, in particular of the beta-1 adrenergic receptor and chemokine receptors such as the C-C chemokine receptor type 4 and 5 [8]. The signal pathways mediated via these receptors are prone to be involved in processes such as chronic allograft vasculopathy and immunoactivation with subsequent rejection events leading to the progressive allograft loss in renal allograft recipients. The single-nucleotide polymorphism, *GNAS* c.393C>T, is a well-studied variant with functional consequences in the gene *GNAS*. Up to now, over 50 studies investigated the role of this polymorphism mainly in neoplasm and arterial hypertension. Besides the *GNAS* c.393C>T polymorphism, somatic missense mutations such as R201H, R201L, R201s, and R201C in the exon 8 of the *GNAS* gene and mutations in the exon 9 such as Q227R and Q227L were identified to have a clinical significance in several tumor entities [8,16,17,18]. These mutations occur in pituitary tumors, appendiceal mucinous adenocarcinoma, duodenal adenocarcinoma, intraductal papillary mucinous neoplasm, and bone tumors [8,24,25,26]. The R201 and Q227 mutations cause constitutive activation of the Gαs, resulting in an overproduction of cAMP and promoting cancer-related signal pathways such as MAPK/ERK [8,24,25,26]. However, for these above-mentioned mutations, the implication in vascular alterations or immunological processing was not examined so far. We hypothesized that increased Gαs-protein signaling due to the *GNAS* c.393C>T polymorphism might negatively affect renal allograft survival, leading to the formation of de novo DSAs against human leukocyte antigens (anti-HLA DSAs), the occurrence of rejection, and increased susceptibility to viral infections.

## 2. Materials and Methods

### 2.1. Study Population

This retrospective single-center study was designed to investigate the relationship between the *GNAS* c.393C>T polymorphism and the clinical outcome after renal transplant. It involved 436 patients who received renal allografts at the University Hospital Essen between January 2011 and December 2015. Three children younger than 18 years and 10 recipients who experienced allograft failure within the first 90 days after transplant were excluded from the analysis. The study was approved by the ethics board (19-9071-BO) of the medical faculty of the University Duisburg-Essen.

Study subjects were followed up for as long as 9 years after renal transplant. Clinical and laboratory data were collected by a review of the electronic medical records. Blood samples used for genotyping were obtained once, before transplant during regular visits for wait listing. All documented rejection episodes were biopsy-proven. Biopsies were performed for cause only during the study period and were analyzed according to the latest available Banff grading criteria [27]. The estimated glomerular filtration rate (eGFR) was calculated with the Chronic Kidney Disease Epidemiology Collaboration (CKD-EPI) equation [28]. Allograft failure was defined as a return to dialysis, and eGFR reduction was defined as a decrease of more than 50% in renal function.

BK polyomavirus (BKV) replication in serum was tested with quantitative real-time polymerase chain reaction (PCR) [29]. DNA was isolated using a commercial kit (QIAamp DNA Mini Kit, cat. no./ID 51306, Qiagen, Hilden, Germany) or via MagNa Pure 24 system (Roche, Basel, Switzerland), following the manufacturer’s instructions. Quantitative BKV DNA detection was done by real-time PCR using the RealStar BKV PCR kit1.0 (Altona Diagnostics, Hamburg, Germany) and the Light Cycler 96 system (Roche, Basel, Switzerland). BK viremia was defined as the presence of more than 400 copies per mL of BKV DNA, which was set as the limit of quantification of the real-time PCR assay. Patients were regularly screened for BK viremia at 1, 3, 6, 12, and 24 months after transplant and then once annually as recommended by the 2009 Kidney Disease: Improving Global Outcomes (KDIGO) guidelines [30]. Additional BK viremia diagnostic tests were performed for patients with allograft dysfunction. High-dose BK viremia was defined as the presence of 10,000 or more BKV DNA copies, in concordance with previous reports suggesting that a BK viral load of 10^4^ copies per mL or more predicted subsequent progression to BKV-associated nephropathy [31]. The diagnosis of BKV-associated nephropathy was made by experienced nephropathologists who examined biopsy specimens of renal allografts according to standard criteria [32].

Cytomegalovirus (CMV) infection was defined by CMV viremia higher than 65 IU/mL, and Epstein–Barr virus (EBV) reactivation was suspected when EB viremia higher than 1000 IU/mL was detected.

Trough levels of immunosuppressive agents, such as tacrolimus and mycophenolate mofetil (MMF), were measured weekly for the first 3 months after transplant, then monthly up to 6 months after transplant, and then at least twice annually. Tacrolimus trough levels were measured with chemiluminescent microparticle immunoassays (CMIA, Architect Tacrolimus, Abbott Diagnostics, Lake Forest, IL, USA), and MMF trough levels were determined with enzyme immunoassays (Siemens, Berlin, Germany). Tacrolimus trough levels of 5 to 7 ng/mL were considered to be target levels for non-viremic patients, whereas lower tacrolimus target trough levels of 3 to 5 ng/ml were preferred for BK-viremic patients.

### 2.2. HLA Typing of Recipients and Donors

For HLA typing of recipients and donors, we isolated DNA from peripheral blood samples. DNA was extracted using spin columns (Qiagen, Hilden, Germany) or by an automated system, using magnetic beads (Chemagen, Baesweiler, Germany). HLA class I (HLA-A, -B, -C) and II (HLA-DRB1, -DQB1) typing was performed at the first-field resolution level, as previously described [33]. Second-field typing was performed to type for selected high-resolution HLA alleles and serological equivalents, according to established Eurotransplant procedures [34]. HLA-DP and HLA-DQA typing was not performed, and HLA-DP- and HLA-DQA-specific antibodies were excluded from further analysis with respect to a putative donor specificity of the anti-HLA-DP and -DQA antibodies.

### 2.3. HLA Antibody Detection and Specification

All patients were screened for anti-HLA class I and II antibodies before transplant. The pretransplant patient sera collected closest to the date of transplant were used for screening. Pretransplant sensitization status was determined for all patients with the standard immunoglobulin G (IgG) complement-dependent cytotoxicity (CDC) test with or without the addition of dithiothreitol (DTT) to exclude antibodies of the IgM isotype. In addition, all patients were tested with a Luminex-based LABScreen mixed bead assay (One Lambda, Thermo Fisher Scientific, Inc., Waltham, MA, USA). In step-by-step analysis [35], the sera that tested positive for anti-HLA class I or II were subsequently specified with LABScreen single antigen bead (SAB) assays (One Lambda, Thermo Fisher Scientific, Inc. Waltham, Massachusetts, USA). All beads with normalized median fluorescence intensity (MFI) values higher than 1000 were considered to be positive for anti-HLA antibodies. To address the potential effect of interfering antibodies or prozone effects on our MFI analyses, we analyzed the sera after multiple freezing and thawing cycles and ethylenediaminetetraacetic acid (EDTA) treatment [36]. 

The results of pretransplant lymphocytotoxic T-cell crossmatches (CDC crossmatch) were negative for all recipients. Anti-HLA antibody status after transplant was monitored at months 3, 6, and 12 after transplant and annually thereafter. Additional screening was performed if allograft dysfunction developed. For the current study, de novo anti-HLA antibodies were determined as early as 4 weeks after renal transplant. We considered samples to be positive for de novo anti-HLA antibodies only when the antibodies were detected at least twice. Nonrecurring evidence of weak anti-HLA antibodies after transplant was assessed as an artifact and therefore was not considered.

### 2.4. GNAS Genotyping

The single-nucleotide polymorphism located in the exon 5 of the *GNAS* gene, *GNAS* c.393C>T (rs 7121), was for the first time detected by Mattera et al., and Kozasa et al. [5,37]. Genomic DNA was extracted from 200 µL blood collected with EDTA with the QIAamp DNA Blood Mini Kit (Qiagen, Hilden, Germany). PCR was performed with 1.8 µL genomic DNA in the concentration of 10 ng/ μL and 30 µL Taq DNA-Polymerase 2× Master Mix Red (Ampliqon, Odense, Denmark) with the following conditions: initial denaturation at 95 °C for 3 min; 38 cycles with denaturation at 95 °C for 30 sec, annealing at 64 °C for 30 s, and elongation at 72 °C for 30 sec each; final elongation at 72 °C for 10 min (forward primer: 5′ TGTGGCCGCCATGAGCAA 3′; reverse primer 3′ TAAGGCCACACAAGTCGGGGT 5′). PCR products were digested with BseGI (Thermo Scientific, Dreireich, Germany), and restriction fragments were analyzed by agarose gel electrophoresis (Figure 1). For the various genotypes, the results of restriction fragment length polymorphism (RFLP)-PCR assays were validated by Sanger sequencing.

Hardy–Weinberg equilibrium (HWE) was calculated with Pearson’s chi square (*χ*^2^) goodness-of-fit test, and samples were considered to deviate from HWE at a significance level of *p* <0.05. Genotypes were compatible with HWE (*χ*^2^ = 2.38; *p* = 0.12).

### 2.5. Statistical Analysis

Categorical variables were expressed as numbers and percentages. Groups were previously tested for normality distribution using the Shapiro–Wilk test. While continuous variables were not normally distributed, the Mann–Whitney test was carried out for the comparison of two groups. Statistically significant differences between groups were verified with the two-tailed *χ*^2^ test. Survival curves were generated by the Kaplan–Meier method and were analyzed with the log-rank test. A multivariate Cox regression analysis was used to test for an independent effect of *GNAS* genotypes on the development of BK viremia or BKV-associated nephropathy. In the multivariate analysis, we adjusted for selected covariables that had been explored in a previous univariate analysis. *p* values were considered to be statistically significant at the level of *p* < 0.05. All data analyses were calculated with GraphPad Prism version 6 (GraphPad Software, Inc., La Jolla, CA, USA) and IBM SPSS Statistics version 23 (IBM Corp., Armonk, NY, USA).

## 3. Results

### 3.1. Baseline Characteristics of the Study Population

The study cohort consisted of 436 adult renal allograft recipients who underwent renal transplant at our center between January 2011 and December 2015. The TC/TT genotypes were detected in 319 (73%) patients, whereas the homozygous CC genotype was detected in 117 (27%) patients. 

As shown in Table 1, patients were classified according to *GNAS* genotypes, and their clinical characteristics were compared. Patients with the CC genotype and those with the TT/TC genotypes were similar in regard to age and sex distribution and cause of renal failure (renal diseases). The entire cohort was treated with calcineurin inhibitors; most patients (92%) were given tacrolimus. Maintenance immunosuppressive therapy contained MMF or mycophenolic acid (MPA) for 362 (83%) patients and mammalian target of rapamycin (mTOR) inhibitors for 73 (17%) patients. No significant differences in maintenance immunosuppression were found between *GNAS* genotypes (Table 1).

In general, 72 (12%) of the 436 renal allograft recipients were undergoing a second transplant. Preformed anti-HLA antibodies were detected in 163 (37%) patients, and preformed anti-HLA DSAs were found in 38 (9%) (Table 1). No difference was found between *GNAS* genotypes with regard to previous transplant or the occurrence of preformed anti-HLA antibodies or anti-HLA DSAs (Table 1). 

### 3.2. Recipients with GNAS CC Genotype Are at Lower Risk of BK Viremia 

BK viremia developed among 101 (23%) of the 436 recipients (Table 2). The frequency of BK viremia was significantly lower among the CC genotype carriers (17 (15%)) than among the TC/TT genotype carriers (84 (26%)); *p* = 0.01; Table 2). The CC genotype was also associated with higher BK viremia-free survival rates (*p* = 0.025; Figure 2A). 

Our analyses of the entire study population found a trend toward worse allograft survival among recipients with BK viremia than among those without (*p* = 0.053; Figure 2B). Interestingly, we found a statistically significant decrease in renal allograft survival rates among T allele carriers with evidence of BK viremia (*p* = 0.002; Figure 2C), but allograft loss due to BK viremia was less frequent among homozygous C allele carriers than among T allele carriers (Figure 2C). 

A univariate analysis showed that deterioration of allograft function, rejection, the use of mTOR inhibitors and MPA/MMF, and in particular the levels of MMF/MPA and tacrolimus at the timepoint of occurrence of BK viremia, along with *GNAS* genotype status, exerted a statistically significant effect as potential covariables influencing the development of BK viremia (Table 3). However, multivariate analysis showed that *GNAS* CC genotype is an independent protective factor against BK viremia (relative risk, 0.54; *p* = 0.04), along with tacrolimus trough levels at the timepoint of viremia, rejection events, and decrease in eGFR (Table 3). 

With respect to other viral and bacterial infection complications during the clinical course after transplant, we found no significant differences between *GNAS* genotypes (Table 2). The *GNAS* CC genotype led to a lower occurrence of Epstein–Barr virus reactivation; however, the difference was not statistically significant (*p* = 0.07; Table 2).

### 3.3. GNAS C393T CC Genotype Is a Protective Factor against BKV-Associated Nephropathy 

Our diagnosis of BKV-associated nephropathy was based on the results obtained from biopsy samples from the renal allograft. Accordingly, we found that 30 (7%) of the recipients exhibited BKV-associated nephropathy. As expected, the development of BKV-associated nephropathy after transplant was related to reduced allograft survival in our cohort (*p* < 0.0001; Figure 3A). 

In agreement with our results for BK viremia, most of the recipients in whom BKV-associated nephropathy developed (27 (8%)) carried at least one T allele, whereas the CC genotypes seemed to be protective (3 (3%); *p* = 0.03; Table 2). We also found that allograft survival rates were lower among *GNAS* T allele carriers than among patients with the CC genotype (*p* = 0.043; Figure 3B).

Next, we investigated the influence of several typical risk factors for BKV reactivation on the development of BKV-associated nephropathy. Except for the *GNAS* CC genotype, decrease in eGFR, rejection events, and higher average trough levels of tacrolimus and MMF as a sign of possible over-immunosuppression, our univariate analysis found no other previously described disadvantageous factors that were significantly associated with the occurrence of BKV-associated nephropathy (Table 4). Multivariate Cox regression confirmed the results of the univariate analysis in terms of the protective effect of the *GNAS* CC genotype against the development of BKV-associated nephropathy (relative risk, 0.27; *p* = 0.036; Table 4). 

### 3.4. High-Dose BK Viremia Is the Main Risk Factor for the Progression to BKV-Associated Nephropathy and Graft Loss among Recipients with BK Viremia 

Among the subgroup of recipients with BK viremia, the progression to BKV-associated nephropathy was attributed to the reduction of allograft function and rapid allograft loss during the follow-up period (*p* < 0.0001; Table 5). Consistent with previous reports, our analysis of patients with BK viremia provided evidence of a statistically significant effect of high-dose BK viremia on the progression to BKV-associated nephropathy (22 of 30 patients (73%) with BKV-associated nephropathy vs. 17 of 71 patients (24%) without BKV-associated nephropathy; *p* = 0.0001; Table 5), whereas patients exhibiting low viral load (≥10,000 copies) did not experience BKV-associated nephropathy (8 of 30 patients (27%) with BKV-associated nephropathy vs. 54 of 71 patients (76%) without BKV-associated nephropathy; *p* = 0.0001). Moreover, high-dose BK viremia was associated with allograft failure more often than was low-dose viremia (14 (82%) vs. 3 (18%); *p* = 0.0001, data not shown). A relevant portion of patients with BK viremia who experienced further progress to BKV-associated nephropathy expressed de novo anti-HLA DSAs before the detection of BK viremia (5 of 30 patients (17%)) than did patients without biopsy-proven BKV-associated nephropathy (3 of 71 patients (4%); *p* = 0.01, Table 5).

There was a slight trend toward a decreased proportion of homozygous C allele carriers among recipients with BKV-associated nephropathy due to BK viremia (3 of 30 patients (10%)) than among recipients without BKV-associated nephropathy (14 of 71 patients (20%); *p* = 0.23; Table 5), although the difference was not statistically significant. Surprisingly, of the 101 recipients with BK viremia, all 17 (20%) recipients who experienced renal allograft failure during the study period carried the T allele (*p* = 0.04). On the other hand, none of the recipients with BKV viremia who carried the CC genotype exhibited subsequent allograft loss.

The proportion of recipients with low-dose BKV viremia was higher among CC genotype carriers (13 of 17 patients (77%)) than among T allele carriers (49 of 84 patients (58%); *p* = 0.16). In contrast, high-dose BK viremia occurred more frequently among T allele carriers (35 of 84 patients (42%)) than among CC genotype carriers (4 of 17 patients (23%); *p* = 0.16). 

### 3.5. No Significant Association between GNAS Genotypes and Rejection Events or Formation of De Novo Anti-HLA DSAs after Renal Transplant

As shown in Table 2, we found no association between *GNAS* genotypes and the occurrence of rejections events, even after stratification according to various Banff categories. Similarly, the rates of allograft failure were comparable between CC genotype carriers and TC/TT genotype carriers. 

Nevertheless, our analyses found a trend toward a lower frequency of de novo anti-HLA DSAs and better de novo anti-HLA DSA-free survival (*p* = 0.055; Appendix A) among homozygous C allele carriers (8 of 117 patients (7%)) than among T allele carriers (43 of 319 patients (13%); *p* = 0.056; Table 2). This trend did not persist in a further multivariate analysis (Appendix A). We were concerned that the association between TC/TT genotypes and de novo anti-HLA DSAs might be biased by the higher proportion of patients with BK viremia among de novo anti-HLA DSA-positive patients who experienced a subsequent reduction of immunosuppressive therapy, a reduction that might consequently result in the development of de novo anti-HLA DSAs in this group of patients. However, the frequency of de novo anti-HLA DSAs was comparable between patients with BK viremia (10 of 101 patients (10%)) and those without it (41 of 335 patients (12%); *p* = 0.52). 

## 4. Discussion

The present study focused on the analysis of the effects of *GNAS* c.393C>T polymorphism on clinical outcome parameters after renal transplant. Most of the renal allograft recipients carried the 393T allele, whereas 117 of the 436 recipients (27%) carried the CC genotype. The main finding of the study was that the CC genotype of *GNAS* exerts a protective effect against the development of BK viremia and the subsequent progression to BKV-associated nephropathy. As expected, BK viremia exerts a negative effect on renal allograft function and survival. Unlike low-dose BK viremia, high-dose BK viremia is associated with rapid progression to BKV-associated nephropathy and with inferior allograft survival. Among recipients with BK viremia, CC genotype carriers tended to have low-dose viremia and none of them experienced allograft loss. 

BK viremia due to donor-derived BKV infection arising from the transplanted kidney itself occurs among 10% to 30% of renal allograft recipients [38,39,40,41]. The frequency of BK viremia (23%) detected in our cohort is in line with previous findings. One of the main reasons for the dysfunction of renal allografts is BK viremia manifesting as BKV-associated nephropathy, found in as many as 10% of renal transplant recipients [39,40]. Thus, the prevalence of BKV-associated nephropathy (7%) documented in our study as leading to a significantly higher rate of renal allograft failure than among recipients without BKV-associated nephropathy is consistent with the findings of a variety of previous reports [31]. 

Recent publications have reported that high-dose BK viremia, as defined by a cutoff level of more than 10^4^ copies/mL, is associated with a higher risk of the detection in biopsy specimens of indicators positive for BKV-associated nephropathy and poor allograft outcome [42,43,44]. On the other hand, no negative effect on renal allograft function and survival has been reported for patients with evidence of low-dose BK viremia. Our observations, including significantly higher rates of BKV-associated nephropathy and allograft dysfunction among patients with high-dose viremia than among those with low-dose viremia, correspond with those of previous reports [42,43,44].

Several risk factors have been proposed to predispose patients to the development of BK viremia and BKV-associated nephropathy [39]: the representative host-derived factors are deceased donation, episodes of acute rejection, and burden of immunosuppressive drugs, such as tacrolimus and MMF [39,40]. Our findings could not confirm the previously described unfavorable effects of male sex, older age, cold ischemia time, previous transplants, deceased donation, delayed graft function, or HLA mismatches on the development of BK viremia and BKV-associated nephropathy. Mengel et al., suggested a strong relationship between the use of a combination of tacrolimus and MMF and the occurrence of BKV-associated nephropathy [45]. Although we found no effect of the use of tacrolimus as a maintenance immunosuppression on the onset of BK viremia and BKV-associated nephropathy, trough levels of tacrolimus were significantly higher, as measured 3 months before the detection of BK viremia or 3 months before the evidence of BKV-associated nephropathy in patients exhibiting BV viremia or BKV-associated nephropathy compared with nonviremic patients or patients without BKV-associated nephropathy. The frequencies of BK viremia and BKV-associated nephropathy did not differ between patients treated with tacrolimus and those treated with cyclosporine A. In agreement with previous findings, our findings showed that average MMF levels detected 3 months before the event (BK-viremia/ BKV-associated nephropathy) were significantly higher in association with the development of BK viremia and BKV-associated nephropathy. As described in recent reports, the use of mTOR inhibitors is considered to be protective [46]. Surprisingly, we found that the initial maintenance immunosuppression regimens for recipients in whom BK viremia developed after transplant were more likely to contain mTOR inhibitors. This observation did not confirm the protective role of mTOR inhibitor therapy on the occurrence of BK viremia in our cohort; however, our multivariate analysis did not detect any significant effect of mTOR inhibitors. Among the 101 recipients with BK viremia, 12 experienced a subsequent switch from MMF to mTOR inhibitors after the diagnosis of BK viremia, although for 11 of these 12 recipients the change in immunosuppressive therapy occurred because of the concomitant development of BKV-associated nephropathy (data not shown).

In addition to immunosuppressive therapy, rejections were another important factor associated with an increased risk of BK viremia and BKV-associated nephropathy among our cohort. This finding agrees with those of several previous studies showing that recognized acute rejection episodes trigger the development of BK viremia and BKV-associated nephropathy [39,44,47]. 

Along with tacrolimus and MMF-based immunosuppression and rejection episodes, we identified that the common *GNAS* c.393C>T polymorphism is a prognostic factor for the development of BK viremia and BKV-associated nephropathy. We found, for the first time, that the CC genotype of *GNAS*, which was detected in 27% of the renal allograft recipients in our cohort, was related to the decreased occurrence of BK viremia and subsequent BKV-associated nephropathy. Multivariate analysis showed that the CC genotype is an independent protective factor against BK viremia and BKV-associated nephropathy. The effect of the *GNAS* c.393C>T polymorphism on renal allograft outcome and posttransplant complications, such as viral infections, has not been previously explored. Until now, numerous reports have described the relationship between *GNAS* genotype and cancer, showing that the TT/TC genotypes exert a protective effect against cancer progression, with improved overall patient survival rates in a variety of cancer entities [8]. Regarding the pathomechanism explaining the effect of the *GNAS* c.393C>T polymorphism on cancer prognosis, it is known so far that altered mRNA folding caused by the T allele seems to lead to increased mRNA expression and to result in increased Gαs production followed by augmented production of the second messenger cAMP [7,8]. Therefore, Gαs overactivation with consequent increased cAMP expression, which is attributed to the TT/TC genotypes, has been suggested to promote proapoptotic effects in cancer. A benefit from elevated Gαs expression probably due to the proapoptotic effects was observed in particular in patients with bladder cancer, colorectal cancer, gastric cancer, and non-small lung cancer [6,7,9,10,11]. On the other hand, an activated Gαs with subsequent increased cAMP production is known to protect against radiation-induced cell death [8]. Hence, CC genotype resulting in reduced Gαs activity was suggested to be associated with better susceptibility to radiochemotherapy than TT/CT genotypes in patients having breast cancer, intrahepatic cholangiocarcinoma, and esophageal cancer [8,12,13,14,15]. Another study found an increased risk of ventricular tachyarrhythmia among patients carrying the T allele [17]. The research group of Frey et al., also found increased *GNAS* mRNA expression and increased Gαs activity in heart specimens obtained from T allele carriers [7]. 

In our research work, we merged the TT and CT genotype, based on the idea that the presence of the T allele of *GNAS* leads to increased expression of Gαs resulting in increased production of cAMP. Frey and colleagues investigated the functional effects of c.393C>T polymorphism in tumor tissues derived from patients suffering from bladder cancer as well as in human adipose tissue and heart tissue [7]. They detected an increased Gαs mRNA expression due to the T allele. A gene–dose effect was observed with highest expression in TT genotypes followed by CT genotypes [7]. A significantly lower expression of Gαs mRNA was measured in CC genotype carriers [7]. 

We can only speculate about the mechanism underlying the relationship between the CC genotype of *GNAS* and the reduced risk of BK viremia and BKV-associated nephropathy among renal transplant patients. We noted that the CC genotype was missing among most of the recipients with BK viremia. Interestingly, CC genotype carriers among the BK viremic recipients had predominantly low-dose viremia and were mostly protected from progression to BKV-associated nephropathy, which occurred mainly among recipients with the TT/TC genotypes. Additionally, none of the CC genotype carriers with BK viremia experienced allograft loss due to BKV-nephropathy, whereas 20% of T allele carriers experienced subsequent allograft failure. Only 8 (8%) of the 101 recipients with BK viremia exhibited de novo anti-HLA DSAs before the development of BK viremia, a finding indicating that the potential over-immunosuppression caused by previous development of de novo anti-HLA DSAs does not explain the observed association between *GNAS* genotypes and BK viremia. Most recipients with BK viremia, in particular those with BKV-associated nephropathy, experienced de novo anti-HLA DSAs after the evidence of BKV in plasma and after the reduction of immunosuppressive maintenance therapy, a reduction that was necessary for clearing the BKV that might have given rise to the development of de novo anti-HLA DSAs, especially among T allele carriers. 

Furthermore, we considered that differences in immunosuppressive treatment between the two genotype groups may have affected our results. However, the distributions of maintenance immunosuppression drugs, such as MMF, mTOR inhibitors, and tacrolimus, as well as tacrolimus and MMF levels, were comparable between the two genotypes. In addition, G-protein-mediated pathways are not involved in the metabolism of tacrolimus or MMF [48,49]. 

We presumed that Gαs activation in T allele carriers might affect the infectivity or clearance of BKV. A linkage between the *GNAS* c.393C>T polymorphism and susceptibility to malaria was observed in a study involving a South Indian population [50]. In this case, plasmodia parasites interacted with stimulatory Gαs on the erythrocyte membrane so that they could invade the cells [50,51]. It was suggested that the increase in stimulatory Gαs caused by the T allele may facilitate the entry of plasmodia into erythrocytes. BKV is a double-stranded DNA virus that enters via binding to the sialic acid of glycoprotein receptors [50,51]. G-protein-mediated pathways and signal transduction have not been found to participate in the entry process of BKV. 

However, G-protein-coupled signal pathways may play a role in the immune response to BKV infection. CD8^+^ and CD4^+^ T cells are involved in the recognition and clearance of BKV [52]. The large T antigen of BKV can stimulate CD8^+^ lymphocytes, whereas Vp1 preferentially activates CD4^+^ T cells [52,53,54]. CD4^+^ T cells exert their antiviral effects on BKV replication by secreting interferon-γ, tumor-necrosis factor-α, and interleukin-2 [54]. The progression of BKV infection to BKV-associated nephropathy is mediated by infiltration and strong inflammation caused by CD8^+^ and CD4^+^ T cells [39]. Gαs has been shown to be necessary for the differentiation and regulation of CD4^+^ T cells. Deletion of *GNAS* from CD4^+^ T cells, accompanied by reduced cAMP levels and protein kinase A-dependent calcium influx, yielded the absence of Th17 and Th1 cells, an absence that led to a reduction of the inflammatory response [55]. Such inhibition of the CD4^+^ T cell-driven immune response to BKV may occur among patients who express the CC genotype, which is associated with lower Gαs and intracellular cAMP levels than those found among T allele carriers who display Gαs activation [7,8]. Downregulation of CD4^+^ T-cell subsets caused by the CC genotype could protect against the initiation of BKV-associated nephropathy. On the other hand, Lee et al., found that the increased polarization of Th2 CD4^+^ T cells caused by the deletion of *GNAS* in the dendritic cells of mice provoked Th2-mediated immunopathologies, such as spontaneous bronchial asthma [56].

We found no relevant association with the *GNAS* genotype in regard to other clinical outcome parameters after renal transplant. The rates of allograft loss and rejection events did not differ significantly between CC carriers and TC/TT carriers. Only a slight trend toward increased numbers of de novo anti-HLA DSAs was detected in T allele carriers, and this trend disappeared after the multivariate analysis was adjusted for covariates. We hypothesized that this observation may be linked to a lower immunosuppressive effect among this subgroup of patients. However, MMF levels and tacrolimus trough levels, measured 1, 3, and 6 months after transplant and during the first 3 years after transplant, did not differ at any time point between the T allele carriers and the homozygous C allele carriers (data not shown). Our findings about de novo anti-HLA DSAs were probably biased by the higher frequency of BK viremia among T allele carriers who might have required a reduction in immunosuppressive treatment. However, it is noteworthy that the rates of de novo anti-HLA DSAs were similar among recipients with and without BK viremia. 

Why the *GNAS* genotype exerted no effect on allograft survival or the occurrence of rejection remains unclear. To our knowledge, no other study has investigated the role of the *GNAS* c.393C>T polymorphism in renal transplants. A study by Beige et al., examined the effect of another polymorphism in the β3-subunit of G proteins, the *GNB3* c.825C>T polymorphism, on renal allograft function [57]. The research group found that, in terms of this G-protein-associated polymorphism, the donor’s genotype rather than the recipient’s genotype was associated with deteriorated allograft survival rates and with chronic rejection among 320 white patients followed up for 3 years. In our present study, donors’ genotypes were not determined. 

We are also aware that our study has certain limitations, primarily its retrospective nature. Although the study involved 436 renal allograft recipients, the number of those with BK viremia, and in particular those with BKV-associated nephropathy, was low. The low number of recipients with BKV-associated nephropathy [31] did not allow us to perform additional multivariate analyses to answer the question of whether the *GNAS* genotype may be a relevant risk factor promoting progression from BK viremia to BKV-associated nephropathy. Additionally, we performed only indication biopsies, not scheduled biopsies of allografts after transplant. Therefore, we could have missed rejection episodes and early stages of BKV-associated nephropathy. Additionally, we could not provide the *GNAS* genotype of the donor. Hence, we may have failed to find a relationship between allograft dysfunction und the occurrence of rejection episodes. Finally, because this was an observational cohort study, our results are descriptive. Additional studies are necessary for elucidating the potential pathomechanism underlying the protective effect of the CC genotype of *GNAS* against BK viremia and BKV-associated nephropathy.

In conclusion, our study, which involved a large cohort of renal allograft recipients, found that the CC genotype of the *GNAS* c.393C>T polymorphism is an independent protective factor against the development of BK viremia and subsequent BKV-associated nephropathy. Screening for the *GNAS* genotype immediately after transplant may provide additional information predicting a favorable prognosis with regard to BK viremia and BKV-associated nephropathy among recipients with the CC genotype. 

## Figures and Tables

**Figure 1 pathogens-11-01138-f001:**
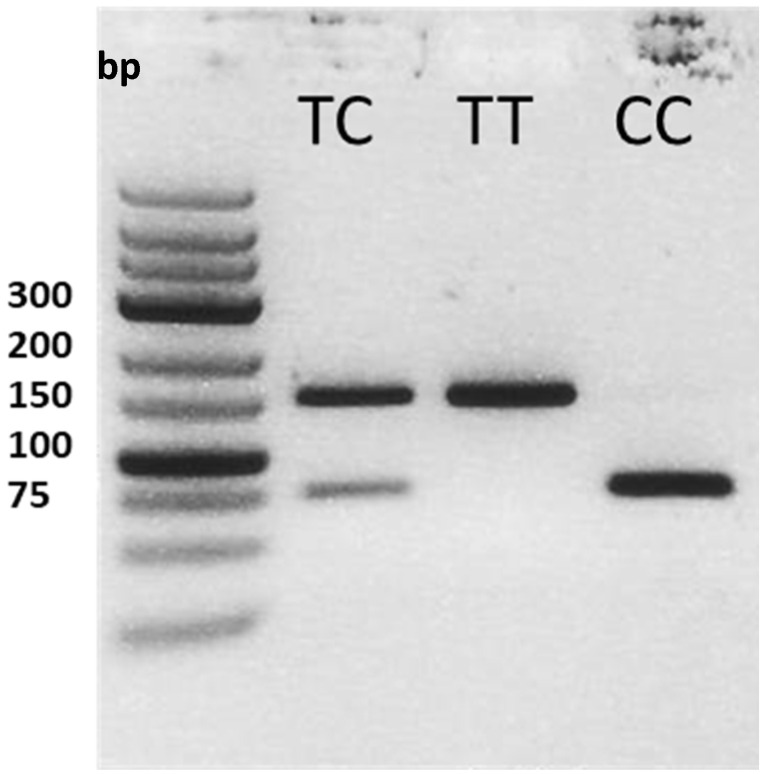
The representative agarose gel showing the amplicon lengths corresponding to the three genotypes of the GNAS (145 bp, TT genotype 145 bp, TC 145 bp, 73 bp and 72 bp, CC 73 and 72 bp).

**Figure 2 pathogens-11-01138-f002:**
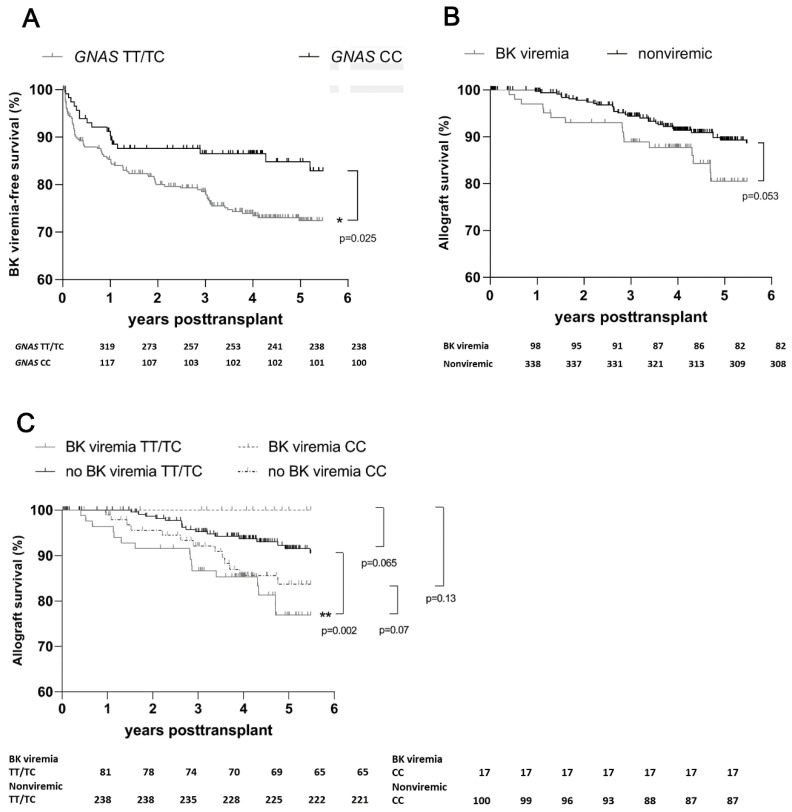
Occurrence of BK viremia according to *GNAS* genotype status and reduced allograft survival due to BK viremia among 436 renal allograft recipients during 5-year follow-up after transplant. (**A**) Graft survival and occurrence of BK viremia according to *GNAS* genotype status (*p* = 0.025). (**B**) Renal allograft survival according to the evidence of BK viremia (*p* = 0.07). (**C**) Renal allograft loss due to BK viremia among recipients with the *GNAS* TT/TC genotypes and those with the CC genotype (*p* = 0.006 vs. *p* = 0.13). *, *p* = 0.05; **, *p* = 0.01. BKV—BK virus.

**Figure 3 pathogens-11-01138-f003:**
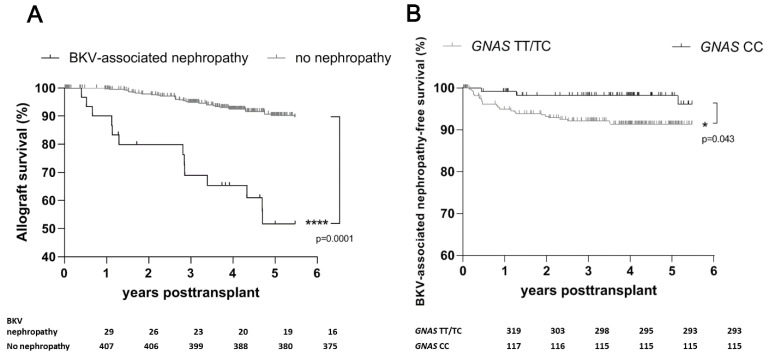
Development of BKV-associated nephropathy in relation to *GNAS* genotype status and accelerated allograft failure due to BKV-associated nephropathy among 436 renal allograft recipients during 5-year follow-up after transplant. (**A**) Renal allograft survival according to the appearance of BKV-associated nephropathy (*p* < 0.0001). (**B**) Graft survival and development of BKV-associated nephropathy according to *GNAS* genotype (*p* = 0.043). *, *p* = 0.05; ****, *p* < 0.0001. BKV—BK virus.

**Table 1 pathogens-11-01138-t001:** Baseline characteristics of 436 renal allograft recipients.

				*χ* ^2^	OR	*p* Value
	All Recipients n = 436	*GNAS* TT/TC n *= 319*	*GNAS* CC n = 117			
**Recipients**						
Age in years, median (range)	53 (18–81)	53 (18–81)	52 (18–79)			0.49
Women, n (%)	183 (42)	132 (41)	51 (44)	0.17	0.91	0.68
Previous transplants, n (%)	52 (12)	41 (13)	11 (9)	0.97	1.42	0.32
CMV status positive, n (%)	270 (62)	204 (64)	66 (56)	2.06	1.37	0.15
CMV high risk (D+/R−), n (%)	75 (17)	51 (16)	24 (21)	1.23	0.74	0.27
PRA, n (%)	37 (8)	24 (8)	13 (11)	1.42	0.65	0.23
Preformed anti-HLA antibodies, n (%)	163 (37)	114 (36)	49 (42)	1.38	0.77	0.24
Class I, n (%)	89 (20)	61 (19)	28 (24)	1.22	0.75	0.27
Class II, n (%)	27 (6)	18 (6)	9 (8)	0.62	0.72	0.43
Classes I and II, n (%)	47 (11)	35 (11)	12 (10)	0.05	1.08	0.83
Preformed anti-HLA DSAs, n (%)	38 (9)	31 (10)	7 (6)	1.50	1.69	0.22
Rest diuresis in ml, median (range)	500 (0–3000)	500 (0–3000)	500 (0–2800)			0.28
Delayed graft function, n (%)	99 (23)	68 (21)	31 (26)	1.31	0.75	0.25
Cold ischemia time in minutes, median (range)	660 (58–1869)	660 (59–1869)	649 (58–1711)			0.67
Warm ischemia time in minutes, median (range)	26 (11–82)	62 (11–82)	25 (11–75)			0.93
**Donor**						
Deceased donors, n (%)	313 (72)	227 (71)	86 (74)	0.23	0.89	0.63
Age in years, median (range)	52 (0–82)	52 (3–82)	51 (0–85)			0.63
Women, n (%)	203 (47)	145 (45)	58 (50)	0.58	0.85	0.45
CMV status, +/−, n (%)	245 (56)	184 (58)	61 (52)	1.07	1.25	0.30
ABO-incompatible transplant, n (%)	33 (8)	26 (8)	7 (6)	0.57	1.39	0.45
**Immunosuppression at transplant**						
IL-2 receptor antagonist, n (%)	405 (93)	298 (93)	107 (91)	0.50	1.33	0.48
ATG, n (%)	23 (5)	14 (4)	9 (8)	1.87	0.55	0.17
Calcineurin inhibitor, n (%)	436 (100)	319 (100)	117 (100)			
Tacrolimus, n (%)	402 (92)	296 (93)	106 (91)	0.57	1.34	0.45
Tacrolimus extended-release formulation, n (%)	26 (6)	19 (6)	7 (6)	0.0001	1	0.99
Cyclosporine A, n (%)	34 (8)	23 (7)	11 (9)	0.57	0.75	0.45
mTOR inhibitor, n (%)	73 (17)	55 (17)	18 (15)	0.21	1.15	0.65
MMF/MPA, n (%)	362 (83)	264 (83)	98 (84)	0.06	0.93	0.80
Steroids, n (%)	436 (100)	319 (100)	117 (100)			
Rituximab, n (%)	5 (1)	4 (1)	1 (1)	0.12	1.47	0.73
Other, n (%)	3 (1)	2 (1)	1 (1)	0.06	0.73	0.80
**HLA mismatches**						
MM (A/B), n (%)	362 (83)	268 (84)	94 (80)	0.82	1.29	0.37
HLA class I MM (A/B): 1–2, n (%)	214 (49)	161 (50)	53 (45)	0.92	1.23	0.34
HLA class I MM (A/B): 3–4, n (%)	148 (34)	107 (34)	41 (35)	0.09	0.94	0.77
MM (DR), n (%)	312 (72)	226 (71)	86 (74)	0.30	0.88	0.59
HLA class II MM (DR): 1, n (%)	206 (47)	145 (45)	61 (52)	1.53	0.77	0.22
HLA class II MM (DR): 2, n (%)	106 (24)	81 (25)	25 (21)	0.75	1.25	0.39
**Causes of renal failure**						
1. Diabetic glomerulosclerosis, n (%)	40 (9)	30 (9)	10 (9)	0.08	1.11	0.78
2. Chronic glomerulonephritis, n (%)	117 (27)	88 (28)	29 (25)	0.34	1.16	0.56
3. Nephrosclerosis, n (%)	58 (13)	44 (14)	14 (12)	0.25	1.18	0.62
4. Polycystic kidney disease, n (%)	66 (15)	45 (14)	21 (18)	0.98	0.75	0.32
5. Tubulointerstitial nephritis, n (%)	15 (3)	11 (3)	4 (3)	0.0002	1.01	0.99
6. Congenital anomalies, n (%)	39 (9)	29 (9)	10 (9)	0.03	1.07	0.86
7. Autoimmune disease, n (%)	18 (4)	11 (3)	7 (6)	1.39	0.56	0.24
8. Amyloidosis, n (%)	4 (1)	4 (1)	0	1.48	1.47	0.22
9. Reflux nephropathy/recurrent pyelonephritis, n (%)	21 (5)	13 (4)	8 (7)	1.43	0.58	0.23
10. HUS, n (%)	8 (2)	6 (2)	2 (2)	0.01	1.10	0.91
11. Other, n (%)	50 (11)	38 (12)	12 (10)	0.23	1.18	0.63

ATG—anti-thymocyte globulin; CMV—cytomegalovirus; D—donor; DSA—donor-specific antibody; HLA—human leukocyte antigen; HUS—hemolytic uremic syndrome; IL-2—interleukin-2; MM—mismatch; MMF—mycophenolate mofetil; MPA—mycophenolic acid; mTOR—mammalian target of rapamycin; OR—odds ratio; PRA—panel-reactive antibodies; R—recipient.

**Table 2 pathogens-11-01138-t002:** Characteristics of renal allograft outcome and infectious complications according to *GNAS* genotypes.

				*χ* ^2^	OR	*p* Value
	All Recipients n = 436	*GNAS* TT/TC n = 319	*GNAS* CC n = 117			
Rejection Banff category 4, n (%)	81 (19)	56 (18)	25 (21)	0.82	0.78	0.36
Rejection Banff category 3, n (%)	77 (18)	62 (19)	15 (13)	2.58	1.64	0.11
Rejection Banff categories 3 and 4, n (%)	137 (31)	103 (32)	34 (29)	0.41	1.16	0.52
ABMR (Banff category 2), n (%)	28 (6)	19 (6)	9 (8)	0.43	0.76	0.51
Rejection Banff categories 2,3 and 4, n (%)	155 (36)	113 (35)	42 (36)	0.01	0.98	0.93
Allograft failure, n (%)	52 (12)	39 (12)	13 (11)	0.10	1.11	0.75
Decrease in eGFR, n (%)	84 (19)	59 (18)	25 (21)	0.45	0.84	0.50
de novo anti-HLA antibodies, n (%)	117 (27)	86 (27)	31 (26)	0.009	1.02	0.92
Class I, n (%)	42 (10)	26 (8)	16 (14)	3.00	0.56	0.08
Class II, n (%)	42 (10)	35 (11)	7 (6)	2.49	1.94	0.12
Classes I and II, n (%)	33 (8)	25 (8)	8 (7)	0.12	1,16	0.73
de novo anti HLA DSAs, n (%)	51 (12)	43 (13)	8 (7)	3.66	2.12	**0.056**
Class I, n (%)	17 (4)	16 (5)	1 (1)	3.96	6.13	**0.05**
Class II, n (%)	24 (6)	21 (7)	3 (3)	2.66	2.68	0.10
Classes I and II, n (%)	10 (2)	6 (2)	4 (3)	0.90	0.54	0.34
**Infections**						
CMV, n (%)	162 (37)	120 (38)	42 (36)	0.11	1.08	0.74
CMV disease, n (%)	34 (8)	21 (7)	13 (11)	2.44	0.56	0.12
BK viremia, n (%)	101 (23)	84 (26)	17 (15)	6.70	2.10	**0.01**
BKV-associated nephropathy, n (%)	30 (7)	27 (8)	3 (3)	4.65	3.51	**0.03**
HEV, n (%)	11 (3)	7 (2)	4 (3)	0.52	0.63	0.47
EBV reactivation, n (%)	84 (19)	68 (21)	16 (14)	3.21	1.71	0.07
Influenza A and B, n (%)	19 (4)	15 (5)	4 (3)	0.34	1.39	0.56
Norovirus, n (%)	9 (2)	5 (2)	4 (3)	1.45	0.45	0.23
HSV, n (%)	6 (1)	5 (2)	1 (1)	0.32	1.85	0.57
VZV/Zoster, n (%)	11 (3)	9 (3)	2 (2)	0.43	1.67	0.51
Pyelonephritis, n (%)	122 (28)	89 (28)	33 (28)	0.004	0.99	0.95
More than 1 episode, n (%)	64 (15)	46 (14)	18 (15)	0.06	0.93	0.8
Pneumonia, n (%)	62 (14)	49 (15)	13 (11)	1.27	1.45	0.26
More than 1 episode, n (%)	21 (5)	16 (5)	5 (4)	0.10	1.18	0.75
Sepsis, n (%)	85 (19)	60 (19)	25 (21)	0.36	0.85	0.55
More than 1 episode, n (%)	21 (5)	13 (4)	8 (7)	1.43	0.58	0.23

ABMR—antibody-mediated rejection; BKV—BK virus; CMV—cytomegalovirus; DSA—donor-specific antibody; EBV—Epstein–Barr virus; eGFR—estimated glomerular filtration rate; HEV—hepatitis E virus; HLA—human leukocyte antigen; HSV—herpes simplex virus; OR—odds ratio; TCMR—T cell-mediated rejection; VZV—Varicella-zoster virus.

**Table 3 pathogens-11-01138-t003:** Results of univariate and multivariate analyses identifying risk factors for and protective factors against occurrence of BK viremia among 436 recipients of renal allografts.

	Recipients with BK Viremia n = 101	Recipients without BK Viremia n = 335	Univariate Relative Risk (95% CI)	*p* Value	Multivariate Relative Risk (95% CI)	*p* Value
**Variable**						
Men, n (%)	65 (64)	188 (56)	1.15 (0.95–1.35)	0.14		
Recipient age in years, median (range)	53 (18–78)	52 (18–81)		0.49		
Previous transplants, n (%)	11 (11)	41 (12)	0.89 (0.47–1.63)	0.71		
Decrease in eGFR, n (%)	28 (28)	56 (17)	1.66 (1.11–2.43)	**0.01**	2.07 (1.18–3.63)	**0.01**
ABO-incompatible transplant, n (%)	6 (6)	27 (8)	0.74 (0.32–1.67)	0.48		
Rejections, n (%)	45 (45)	110 (33)	1.36 (1.03–1.75)	**0.03**	1.20 (0.71–2.03)	0.49
de novo anti-HLA DSAs, n (%)	10 (10)	41 (12)	0.81 (0.42–1.52)	0.52		
de novo anti-HLA, n (%)	20 (20)	97 (29)	0.68 (0.44–1.03)	0.07		
Delayed graft function, n (%)	19 (19)	80 (24)	0.79 (0.50–1.21)	0.29		
Cold ischemia time in minutes, mean (range)	597 (70–1592)	618 (58–1869)		0.37		
Deceased donation, n (%)	72 (71)	241 (72)	0.99 (0.85–1.13)	0.90		
MM (A/B), n (%)	84 (83)	278 (83)	1.00 (0.89–1.10)	0.97		
MM (DR), n (%)	75 (74)	237 (71)	1.05 (0.91–1.19)	0.49		
*GNAS* CC genotype, n (%)	17 (17)	100 (30)	0.55 (0.34–0.87)	**0.01**	0.54 (0.30–0.97)	**0.04**
Induction with rituximab, n (%)	2 (2)	3 (1)	2.21 (0.44–10.88)	0.37		
Induction with ATG, n (%)	8 (8)	15 (4)	1.77 (0.78–3.93)	0.17		
Administration of cyclosporine A, n (%)	8 (8)	26 (8)	1.02 (0.48–2.12)	0.96		
Administration of tacrolimus, n (%)	93 (92)	309 (92)	1.00 (0.92–1.06)	0.96		
Administration of tacrolimus extended-release formulation, n (%)	6 (6)	20 (6)	1.00 (0.42–2.32)	0.99		
Tacrolimus trough level, mean (range)	7.1 (0.8–14.9)	6.2 (0.2–19.3)		**0.0002**	1.07 (1.00–1.15)	**0.04**
Administration of mTOR inhibitors, n (%)	25 (25)	48 (14)	1.73 (1.12–2.62)	**0.01**	0.60 (0.08–4.48)	0.62
Administration of MMF/MPA, n (%)	76 (76)	286 (85)	0.88 (0.77–0.98)	**0.02**	0.52 (0.06–4.46)	0.55
MMF level, mean (range)	4.1 (0.1–18.0)	3.1 (0.0–22.0)		**0.005**	1.05 (0.99–1.11)	0.101

ATG—anti-thymocyte globulin; CI—confidence interval; DSA—donor-specific antibody; eGFR—estimated glomerular filtration rate; HLA—human leukocyte antigen; MM—mismatch; MMF—mycophenolate mofetil; MPA—mycophenolic acid; mTOR—mammalian target of rapamycin.

**Table 4 pathogens-11-01138-t004:** Results of univariate and multivariate analyses identifying risk factors and protective factors for development of BKV-associated nephropathy among 436 patients after renal allograft transplant.

	Recipients with BKV-Associated Nephropathy n = 30	Recipients without BKV-Associated Nephropathy n = 406	Univariate Relative Risk (95% CI)	*p* Value	Multivariate Relative Risk (95% CI)	*p* Value
**Variable**						
Men, n (%)	19 (63)	234 (58)	1.1 (0.78–1.38)	0.54		
Recipient age in years, median (range)	53 (18–76)	53 (18–81)		0.95		
Previous transplants, n (%)	5 (17)	47 (12)	1.44 (0.61–3.07)	0.41		
Decrease in eGFR, n (%)	19 (63)	65 (16)	3.96 (2.68–5.46)	**<0.0001**	10.84 (4.17–28.22)	**<0.001**
ABO-incompatible transplant, n (%)	0	33 (8)	0 (0–1.41)	0.1		
Rejections, n (%)	17 (57)	138 (34)	1.67 (1.13–2.23)	**0.01**	0.94 (0.37–2.37)	0.89
de novo anti-HLA DSAs, n (%)	5 (17)	46 (11)	1.47 (0.62–3.14)	0.38		
de novo anti-HLA, n (%)	8 (27)	109 (27)	0.99 (0.52–1.71)	0.98		
Delayed graft function, n (%)	5 (17)	94 (23)	0.72 (0.31–1.49)	0.41		
Cold ischemia time in minutes, mean (range)	744 (70–1592)	605 (58–1869)		0.31		
Deceased donation, n (%)	25 (83)	288 (71)	1.18 (0.93–1.34)	0.15		
MM (A/B), n (%)	26 (87)	336 (83)	1.05 (0.85–1.16)	0.58		
MM (DR), n (%)	23 (77)	289 (71)	1.08 (0.83–1.26)	0.52		
*GNAS* CC genotype, n (%)	3 (10)	114 (28)	0.36 (0.12–0.93)	**0.03**	0.27 (0.08–0.92)	**0.036**
Induction with rituximab, n (%)	0	5 (1)	0 (0–9.69)	0.54		
Induction with ATG, n (%)	2 (7)	21 (5)	1.29 (0.34–4.45)	0.72		
Administration of cyclosporine A, n (%)	0	34 (8)	0 (0–1.37)	0.10		
Administration of tacrolimus, n (%)	30 (100)	372 (92)	1.09 (0.97–2.53)	0.10		
Administration of tacrolimus extended-release formulation, n (%)	0	26 (6)	0 (0–1.8)	0.15		
Tacrolimus trough level, mean (range)	7.0 (3.0–12.6)	6.2 (0.2–19.3)		**0.04**	1.03 (0.92–1.16)	0.59
Use of mTOR inhibitors, n (%)	7 (23)	66 (16)	1.44 (0.70–2.65)	0.32		
Use of MMF/MPA, n (%)	23 (77)	339 (83)	0.92 (0.71–1.07)	0.34		
MMF level, mean (range)	4.2 (0.1–10.8)	3.1 (0.0–22.0)		**0.04**	1.07 (0.98–1.17)	0.16

ATG—anti-thymocyte globulin; BKV—BK virus; CI—confidence interval; DSA—donor-specific antibody; eGFR—estimated glomerular filtration rate; HLA—human leukocyte antigen; MM—mismatch; MMF—mycophenolate mofetil; MPA—mycophenolic acid; mTOR—mammalian target of rapamycin.

**Table 5 pathogens-11-01138-t005:** Results of analysis of renal transplant outcome and potential risk factors for progression to BKV-associated nephropathy in a subgroup of 101 recipients with BK viremia.

	Patients with BK Viremia n = 101	Patients with BKV-Associated Nephropathy n = 30	Patients without BKV-Associated Nephropathy n = 71	*χ* ^2^	OR	*p* Value
Decrease in eGFR, n (%)	28 (28)	19 (63)	9 (13)	27.0	11.9	**0.0001**
Transplant failure, n (%)	17 (17)	13 (43)	4 (6)	21.4	12.8	**0.0001**
High-dose viremia (≥10E4), n (%)	39 (39)	22 (73)	17 (24)	21.7	8.7	**0.0001**
de novo anti-HLA DSAs, n (%)	10 (10)	5 (17)	5 (7)	2.2	2.6	0.14
Appearance of de novo anti-HLA DSAs before BK viremia, n (%)	8 (8)	5 (17)	3 (4)	6.2	6.8	**0.01**
*GNAS* CC genotype, n (%)	17 (17)	3(10)	14 (20)	1.4	0.45	0.23
Additional increase in viral load at 6-month follow-up, n (%)	10 (10)	4 (13)	6 (8)	0.6	1.7	0.5
mTOR inhibitor as maintenance immunosuppressant, n (%)	26 (26)	6 (20)	20 (28)	0.7	0.6	0.4

BKV—BK virus; DSA—donor-specific antibody; eGFR—estimated glomerular filtration rate; HLA—human leukocyte antigen; mTOR—mammalian target of rapamycin; OR—odds ratio.

## Data Availability

The data presented in this study are available on request from the corresponding author. The data are not publicly available due to ethical restrictions.

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
