# Peer review of "CC Genotype of GNAS c.393C>T (rs7121) Polymorphism Has a Protective Effect against Development of BK Viremia and BKV-Associated Nephropathy after Renal Transplant"

_pathogens, 2022, doi:10.3390/pathogens11101138_

Round 1
Reviewer 1 Report
The authors investigate an association of a polymorphism in the GNAS gene with viremia after kidney transplantation. The manuscript is well written and a genetic association study on the subject is of great interest to the readership. However, a few points need to be addressed to make the conclusion clearer to the readership. In particular, it must be stated why exactly this polymorphism was chosen, as some other polymorphisms have been described in the GNAS gene. Also, it is not clear to this reviewer why the CT genotypes were examined together with TT genotypes and compared to CC genotypes. These issues need to be addressed in the revised version.
Specific comments:
Page 2: line 49 –The T allele shows different effect in various kinds of tumor (eg. better survival in renal, bladder cancer and melanoma as well as different kinds of head and neck cancer), while the CC genotype was associated with worse outcome in some other kinds of cancer. The original papers should be mentioned and cited with regard to different behavior in selected tumor types.
Page 4: Line 153: Can you explain why genotypes were validated by Sanger sequencing? Where was the sequencing done?
Page 4 line 158: Please explain and indicate the tests for normal distribution. What tests were used for normally and not normally distributed linear variables?
Discussion. Page 14 Line 394. This sentence contradicts statements from the introduction that indicate a protective role of CC genotypes in cancer progression. As stated above, the authors must correctly assess and cite the original literature and bring their results to a coherent conclusion
Reviewer 2 Report
This is an interesting study regarding the association of a gene polymorphism with polyomavirus viremia, namely the GNAS gene. The sample number and the statistics seem to support the main conclusions. However the study misses a lot of necessary information concerning the methodological approach. For instance, the authors do not explain the restriction profiles, nor provide the first studies who described this mutation. A table with the restriction profiles, amplicon length, as well as a figure of the agarose gel indicating the heterozugous and homozygous genotypes has to be added.
The authors should also address the following detailed comments:
Lines 12-13: The GNAS gene encodes …. (Gαs) in which organisms? In mammals? In fungi? In Tunicata? Please be specific. The sentence should also be separated in two: the first for the gene and the second for the SNP
Please provide also the results for CT and TT genotypes in the abstract
Please provide details for the BK polyomavirus (BKV) qPCR, as well as for the quantification method regarding the 400 copies per mL
Lines 112-113: Which method / kit was used for DNA isolation?
Lines 145-146: 1.8 μl genomic DNA of what concentration?
Lines 149-150: Did the authors design the primers? If so they should provide detailed information. If not, they should provide a reference
The restriction profiles are missing. How were the genotypes characterised?
Round 2
Reviewer 2 Report
I would only recommend to add the agarose gel image within the text. Regarding all other comments I feel satisfied and the manuscript may be accepted
Author Response
As suggested, we included the representative agarose gel as Figure 1 into the manuscript.
